# Synthesis of Network Polymers by Means of Addition Reactions of Multifunctional-Amine and Poly(ethylene glycol) Diglycidyl Ether or Diacrylate Compounds

**DOI:** 10.3390/polym12092047

**Published:** 2020-09-08

**Authors:** Naofumi Naga, Mitsusuke Sato, Kensuke Mori, Hassan Nageh, Tamaki Nakano

**Affiliations:** 1Department of Applied Chemistry, College of Engineering, Shibaura Institute of Technology, 3-7-5 Toyosu, Koto-ku, Tokyo 135-8548, Japan; ad12097@shibaura-it.ac.jp; 2Graduate School of Science & Engineering, Shibaura Institute of Technology, 3-7-5 Toyosu, Koto-ku, Tokyo 135-8548, Japan; mc18018@shibaura-it.ac.jp; 3Institute for Catalysis and Graduate School of Chemical Sciences and Engineering, Hokkaido University, N 21, W 10, Kita-ku Sapporo 001-0021, Japan; science_as2000@yahoo.com (H.N.); tamaki.nakano@cat.hokudai.ac.jp (T.N.); 4Integrated Research Consortium on Chemical Sciences, Institute for Catalysis, Hokkaido University, N 21, W 10, Kita-ku Sapporo 001-0021, Japan

**Keywords:** gel, porous polymer, multi-functional amine, poly(ethylene glycol), network structure, mechanical property

## Abstract

Addition reactions of multi-functional amine, polyethylene imine (PEI) or diethylenetriamine (DETA), and poly(ethylene glycol) diglycidyl ether (PEGDE) or poly(ethylene glycol) diacrylate (PEGDA), have been investigated to obtain network polymers in H_2_O, dimethyl sulfoxide (DMSO), and ethanol (EtOH). Ring opening addition reaction of the multi-functional amine and PEGDE in H_2_O at room temperature or in DMSO at 90 °C using triphenylphosphine as a catalyst yielded gels. Aza-Michael addition reaction of the multi-functional amine and PEGDA in DMSO or EtOH at room temperature also yielded corresponding gels. Compression test of the gels obtained with PEI showed higher Young’s modulus than those with DETA. The reactions of the multi-functional amine and low molecular weight PEGDA in EtOH under the specific conditions yielded porous polymers induced by phase separation during the network formation. The morphology of the porous polymers could be controlled by the reaction conditions, especially monomer concentration and feed ratio of the multi-functional amine to PEGDA of the reaction system. The porous structure was formed by connected spheres or a co-continuous monolithic structure. The porous polymers were unbreakable by compression, and their Young’s modulus increased with the increase in the monomer concentration of the reaction systems. The porous polymers absorbed various solvents derived from high affinity between the polyethylene glycol units in the network structure and the solvents.

## 1. Introduction

Network structure, such as molecular and geometric structures, in chemically synthesized gels strongly affects their properties. Polymerization (or copolymerization) of bi-functional (or multi-functional) monomers and crosslinking of liner pre-polymers using chemical reactions are widely used to synthesize polymer network in chemical gels. We have been developing several types of gels synthesized by addition reactions between multi-functional monomers as “joint unit” sources and α,ω-bifunctional monomers as “linker unit” sources in some solvents, based on the joint and linker concept. In our previous studies, addition reactions of multi-functional cyclic siloxane or cubic silsesquioxane, thiol, polyol, and acrylate as the multi-functional monomers and α,ω-diolefins, dithiol, diazide, divinyl ether, and diamine compounds as the bi-functional monomers successfully yielded the gels [1,2,3]. The gels formed a homogeneous network structure with extremely narrow mesh size distribution, and their mesh size could be controlled by the molecular length of the bi-functional monomers. However, those gels were too fragile to evaluate their mechanical properties. We have been trying to synthesize the gels with high mechanical properties using conventional addition reactions of generic chemicals and solvents. Various hydrogels with high mechanical properties have been developed by polymerization of functionalized poly(ethylene glycol) (PEG) [4,5,6,7,8,9,10,11,12,13,14,15,16,17,18,19,20,21,22,23]. The hydrogels with PEG moiety are expected to be applied as a biomaterial. The PEG linkage shows high affinity with various organic solvents, and the corresponding gels should be synthesized in organic solvents, so-called organogels. The PEG unit results in good ionic conductivity for cation carriers. The organogels with the PEG unit should be usable for structure material ionic conducting gels. Prior to the development of the ionic conducting gels, we investigated a basic study of the molecular design of organogels with the PEG unit based on the joint and linker concept. In addition, we expected to prepare the porous polymers by means of polymerization induced phase separation by control of affinity between the polymer network and the solvent in the reaction system. We selected bi-functional PEG as the linker monomer. For example, the gels were synthesized by addition reactions of a primary tri-amine, tris(2-aminoethyl)amine (TAEA), as the multi-functional monomer, and polyethylene glycol diglycidyl ether (PEGDE) or polyethylene glycol diacrylate (PEGDA) as the bi-functional monomer in some organic solvents [24]. The reactions yielded the gels with high mechanical properties. We also found the reaction of TAEA and low molecular weight PEGDA in ethanol (EtOH) yielded porous polymers induced by phase separation during the network formation. These polymers showed features derived from the PEG unit in the network structure. Although TAEA is available as a research reagent from some chemical companies, it is not suitable for applications in large scale from industrial perspectives of availability and cost. As the next step, we focus on commercial availability and variation of molecular structure of the multi-functional amine monomer, which can react with PEGDE or PEGDA. We select polyethyleneimine (PEI) as a multi-functional monomer. PEI has primary, secondary, and tertiary amines with branched structure. Diethylenetriamine (DETA) was also used as a corresponding small molecular amine monomer. These chemicals are industrially produced and widely used as washing (water-treatment) agents, adhesives, cosmetics, cultivation, treatment of fiber and paper, chelate, coatings, ion-exchange resins, epoxy resin curing agent, and so on.

In this paper, we describe the synthesis of gels by addition reaction of PEI or DETA, as the multi-functional monomer, and PEGDE using the ring opening addition reaction [25] (Scheme 1) or PEGDA using the aza-Michael addition reaction [26] (Scheme 2). We study the effect of the network structure, feed ratio of amine monomer to PEGDE or PEGDA, and features of the solvents (H_2_O, dimethyl sulfoxide (DMSO), EtOH) on the mechanical properties of the resultant gels. The porous polymers were also obtained in the reactions of PEI or DETA with low molecular weight PEGDA in EtOH, and the structure and properties of the porous polymers were also investigated.

## 2. Materials and Methods

### 2.1. Materials

PEI (Polyethylenimine300) was commercially obtained from Junsei Chemical Co., Ltd. (Tokyo, Japan), and used as received. The structure of PEI is as follows: molecular weight: 300, amine value: 21 mmol/g, [primary amine]/[secondary amine]/[tertiary amine] = 45%:35%:20%. DETA was commercially obtained from Tokyo Chemical Industry Co., Ltd. (Tokyo, Japan), and used without further purification. PEGDE samples, PEGDE400 (epoxy: 263 g/eq.) and PEGDE1000 (epoxy: 574 g/eq.), were kindly donated from NOF CORPORATION (Tokyo, Japan), and were used without further purification. PEGDA samples, PEGDA200, PEGDA400, PEGDA600, and PEGDA1000, were kindly donated from Shin-Nakamura Chemical Co., Ltd. (Wakayama, Japan), and were purified by passing through Al_2_O_3_ column before use to remove polymerization inhibitors. High purity grade DMSO and EtOH were commercially obtained from Kanto Chemical Co., Inc. (Tokyo, Japan), and used as received. Triphenylphosphine (PPh_3_) (Kanto Chemical Co., Inc., Tokyo, Japan) was commercially obtained, and was dissolved in the solvent of the reaction system before use.

### 2.2. Synthesis of Network Polymers

DETA has two primary amines, NH_2_ group, and one secondary amine, NH group. That means five active hydrogens or three amines in one DETA molecule. The reactions of DETA and PEGDE or PEGDA were conducted under the feed ratio of Case 1: [N] (mole of amines in DETA) = [epoxy or acrylate]) or Case 2: [H] (mole of active hydrogens in DETA) = [epoxy or acrylate]), as shown in Scheme 3. In the same way, the reactions of PEI-PEGDE and PEI-PEGDA (Case 1: [N] (mole of amines in PEI) = [acrylate or acrylate], Case 2: [H] (mole of active hydrogens in PEI) = [acrylate or acrylate]) were defined as Case 1 or Case 2, respectively.

#### Synthesis of Gels

A reaction of PEI with PEGDE400 (monomer concentration: 30 wt %, Case 1) is described as a reference. PEI (0.15 g, [N] 2.5 mmol), PEGDE400 (0.67 g, epoxy 2.5 mmol), and DMSO (0.92 mL) were added to a 20 mL vial and stirred by vortex mixer for several minutes to make a homogeneous solution. DMSO solution of PPh_3_ (0.84 mL, 0.076 mM, 2.5 mol% to epoxy group) was added to the reaction solution. The reaction solution was introduced to an ampoule of 10 mL. After the ampoule was sealed, the reaction system was kept at 90 °C for 24 h. The reaction with PEGDE1000 or DETA was conducted under the same procedures. The reactions in H_2_O were conducted at room temperature without PPh_3_.

A reaction of PEI with PEGDA400 (monomer concentration: 30 wt %, Case 1) in DMSO is described as a reference. PEI (0.15 g, [N] 2.5 mmol), PEGDA400 (0.635 g, epoxy 2.5 mmol), and DMSO (1.67 mL) were added to a 20 mL vial, and stirred by vortex mixer for several minutes to make a homogeneous solution. The reaction solution was introduced to an ampoule of 10 mL. After the ampoule was sealed, the reaction system was kept at room temperature for 24 h. The reaction with PEGDA200, PEGDA600, PEGDA1000, DETA, or in EtOH was conducted under the same procedures.

### 2.3. Analytical Procedures

Viscosity of the reaction systems was traced by a tuning fork vibration viscometer, SV-1A (A&D Company Limited, Tokyo, Japan), equipped with a block heater. The measurements were conducted at room temperature or 90 °C.

FT-IR spectra of reaction solutions and gels were recorded on a Jasco FT/IR-410 (JASCO Corporation, Tokyo, Japan). The samples were put between KBr-Real Crystal IR-Card and Slip (International Crystal Laboratories, NJ, USA), and 30 scans were accumulated from 4000 to 500 cm^−1^.

The mechanical properties of the gels or porous polymers were investigated by the compression test using Tensilon RTE-1210 (ORIENTEC Co. LTD., Tokyo, Japan). The test samples, wet gels as prepared state or dried porous polymers, were cut to 0.7–1 cm cubes, and pressed at a rate of 0.5 mm/min at room temperature. Compression test of the gels was conducted at 24 h after the gel formation.

Scanning electron microscopy (SEM) images of the porous polymers were acquired by a JEOL (Tokyo, Japan) JSM-7610F microscope with a lower secondary electron image detector at an acceleration voltage of 3.0 kV.

The surface area of the porous polymers was measured by nitrogen sorption using an Autosorb 6AG (Quantachrome, FL, USA), and determined by the Brunauer–Emmett–Teller (BET) equation.

## 3. Results and Discussion

### 3.1. Sythessis of Network Polymers by Means of Ring Opening Addition Reaction of PEI or DETA and PEGDE

Ring-opening addition reaction of PEI or DETA and PEGDE was conducted in H_2_O, DMSO, or EtOH. The reactions in H_2_O successfully produced the gels at room temperature without catalyst. The reactions at relatively low temperatures and/or without catalyst did not gel in DMSO. The corresponding reactions in the presence of PPh_3_ as the catalyst at 90 °C successfully yielded the gels. The reaction systems in EtOH did not gel despite the presence of the catalyst. The reactions must be conducted at the temperature less than the boiling point of EtOH (60 °C). The reactions under 60 °C in EtOH would not sufficient to form the gels.

The reaction in DMSO was traced by FT-IR spectroscopy. Figure 1 shows the FT-IR spectra of the PEI-PEGDE400 reaction system, monomer concentration: 30 wt % in DMSO. Intensity of absorption peaks at about 800 cm^−1^ (Figure 1 (i)) derived from epoxy group and at about 1150 cm^−1^ (Figure 1 (ii)) and 1600 cm^−1^ (Figure 1 (iii)) derived from amine groups almost disappeared in the spectrum of the gel, indicating the addition reaction of epoxy group with amine would progress successfully.

The viscosity of the PEI or DETA-PEGDE reaction systems, monomer concentration: 30 wt% in H_2_O, was traced by the tuning fork vibration viscometer at room temperature (Appendix A). An inflection point of the profile was defined as the gel formation time (not theoretical gelation time), summarized in Table 1. The gel formation time of the PEI-PEGDE400 reaction system (run 1) was shorter than that of the PEI-PEGDE1000 system (run 5) owing to the higher epoxy (amine) concentration in the reaction system with the same monomer concentration. The PEI-PEGDE400 reaction system of Case 2 (run 2) showed longer gel formation time than that of Case 1 (run 1). The FT-IR spectra of PEI-PEGDE400 gel obtained in the reaction system of Case 2 (Appendix A) showed a similar profile to that of the corresponding gel obtained in the reaction systems of Case 1 (Figure 1). A comparison of these FT-IR spectra indicates that there was not much difference in reaction conversions of these gels. In the case of the reaction of Case 2, addition reaction of primary amine with epoxy would proceed in two steps, as reported in the curing process of diamine and bisphenol A type diepoxy [27]. The first step is the reaction of a primary amine and an epoxy, which forms a secondary amine. After completion of the first step reaction, the secondary amine reacts with another epoxy. The longer gel formation time of Case 2 would be caused by the two steps’ reactions. Steric hindered secondary amine formed by a reaction of primary amine and one epoxy is also possible. The reaction system of PEI-PEGDE400 (run 1) caused the gel to form faster than that of DETA-PEGDE400 (run 3). The concentration of epoxy group (0.96 mol/L) of the PEI-PEGDE400 reaction system was almost same as that of the DETA-PEGDE400 reaction system (1.02 mol/L). One explanation of the result is that the inter-penetration of the polymer networks derived from high molecular weight and highly functionalized PEI, which should play a role of a physical crosslinking point, would occur during the reaction, and attained a short gel formation time. The viscosity of the PEI-PEGDE400 reaction system in DMSO at 90 °C was also traced. The gel formation times of the reaction systems in DMSO were longer than those in the corresponding reaction systems in H_2_O. Low basicity of amines of PEI in low polar solvent of DMSO may lower the reaction rate, which should cause longer gel formation times in DMSO.

The mechanical properties of the PEI or DETA-PEGDE gels were investigated by compression test. Figure 2 shows the stress–strain curves of PEI or DETA-PEGDE400 gels, monomer concentration: 30 wt % in H_2_O, and the results are summarized in Table 2. The gels obtained in the reaction of Case 2 showed higher Young’s modulus than the corresponding gels obtained in the reaction of Case 1. One explanation of the result is that the higher epoxy concentration in the reaction systems of Case 2 should yield gels with high crosslinking density. The gels with PEI showed hard and brittle features in comparison with those with DETA. The result can be explained by the inter-penetration of the polymer networks derived from the specific structure of PEI, as described above. The PEI-PEGDE1000 gels showed soft and flexible features in comparison with the PEI-PEGDE400 gels owing to the lower crosslinking density, derived from the low epoxy concentration in the reaction systems. The PEI-PEGDE400 gels prepared in DMSO showed lower Young’s modulus and higher strain at break than the corresponding gels prepared in H_2_O. The result can be explained by the high affinity between the polymer network and DMSO, which should make the gels soft and flexible.

### 3.2. Synthesis of Network Polymers by Means of Aza-Michael Addition Reaction of DETA or PEI and PEGDA

Aza-Michael addition reaction of PEI or DETA and PEGDA was conducted in H_2_O, DMSO, or EtOH at room temperature. The reaction systems in H_2_O once formed gels, and then turned them into solution owing to the hydrolytic degradation of ester group of PEGDA. The reactions in DMSO successfully yielded the gels. The molecular weight of PEGDA affected the production state of the reaction in EtOH. The reactions with relatively high molecular weight, PEGDA, PEGDA400, PEGDA600, and PEGDA1000, in EtOH yielded the gels. By contrast, the reactions with low molecular weight, PEGDA and PEGDA200, in EtOH formed some states, gel, precipitated polymer, or porous polymer, depended on the reaction conditions. We shall return to this point later.

Figure 3 shows FT-IR spectra of the PEI-PEDA400 reaction system, monomer concentration: 30 wt% in DMSO. The intensity of the absorption peaks at about 700 cm^−1^ (Figure 3 (i)) derived from acrylate and 1150 cm^−1^ (Figure 3 (ii)) and 1600 cm^−1^ (Figure 3 (iii)) derived from amine groups was decreased (almost disappeared) in the spectrum of the gel owing to the addition reaction of the acrylate group with amine.

The viscosity of the PEI-PEGDA reaction systems, monomer concentration: 30 wt% in DMSO, was traced at room temperature to estimate the gel formation time (Appendix A), summarized in Table 2. The gel formation time of the reaction systems of Case 1 increased with the increase in the molecular weight of PEGDA. The concentration of acrylate group of PEGDA in the reaction systems increased with the decrease in the molecular weight of PEGDA under the same monomer concentration, as summarized in Table 2. The higher molar concentration of the acrylate (and amine) group in the reaction system with low molecular weight PEGDA200 yielded the polymer network with high crosslinking density, which should cause the short gel formation time in the homogeneous phase. The gel formation time of the PEI-PEGDA400 reaction system of Case 1 (5 min) was much shorter than that of Case 2 (135 min). The FT-IR spectrum of the PEI-PEGDA400 gel obtained from the reaction system of Case 2 (Appendix A) showed a similar profile to that of the gel obtained from the reaction system of Case 1 (Figure 3). The conversions of acrylate group in the gels of Case 1 and Case 2 evaluated from peak intensity of (i) in Figure 3 and Appendix A based on the intensity of unchanged peak (at 965 cm^−1^) were about 62% and 56%, respectively. Although the conversions of acrylate groups in the gel of Case 2 were a little smaller than that of Case 1, this would not necessarily mean a difference in the reaction rates of Case 1 and Case 2. The longer gel formation time of the PEI-PEGDA400 reaction system of Case 2 can be explained by steric hindered proton transfer of the secondary amine formed by a reaction of a primary amine and an acrylate, as reported in a computational study of an aza-Michael addition reaction [28]. The gel formation times of the DETA-PEGDA400 reaction systems were longer than the corresponding PEI-PEGDA400 reaction systems despite the similar acrylate concentration, as observed in the reactions of PEI or DETA-PEGDE systems. The results can be explained by the inter-penetration of the polymer networks derived from PEI, which would be formed during the reaction.

The gel formation times of the reaction systems in EtOH were also determined (Appendix A), and the results are summarized in Table 2. The corresponding reactions in DMSO and EtOH showed similar gel formation times.

The mechanical properties of the PEI or DETA-PEGDA gels in DMSO were investigated by compression test. Figure 4 shows the stress–strain curves of PEI or DETA-PEGDA400 gels, monomer concentration: 30 wt % in DMSO, and the results are summarized in Table 2. The PEI or DETA-PEGDE400 gels obtained in the reactions of Case 1 showed a higher Young’s modulus than the corresponding gels of Case 2. These results are opposite to those of the PEI or DETA-PEGDE400 gels described above. One explanation of the results is the lower reaction conversions in the reaction systems of Case 2 with PEGDA400, as observed in FT-IR spectroscopy (Appendix A). The reaction systems with low reaction conversion should yield gels with low crosslinking density and soft features. The gels with PEI showed hard and brittle features in comparison with those with DETA, as observed in the PEI or DETA-PEGDE400 gels. The result can be explained by the inter-penetration of the polymer networks derived from the specific structure of PEI, as described above. The PEI-PEGDA600 and PEGDA1000 gels showed soft and flexible features in comparison with the PEI-PEGDA400 gel owing to the lower crosslinking density derived from lower acrylate concentration in the reaction system.

The mechanical properties of the PEI or DETA-PEGDA gels in EtOH were also investigated by the compression test, and the results are summarized in Table 3. The gels in EtOH showed a much lower Young’s modulus than the corresponding gels in DMSO. One explanation of the results is the strict difference in the phase of the gels. The reactions in DMSO occurred in the homogeneous phase, and yielded transparent gels. By contrast, the gels prepared in EtOH slightly turned white. This would be an intermediate state between homogeneous gel and porous polymer induced by phase separation. The phase separation occurred in the reactions with low molecular weight PEGDA200 in EtOH, as described below. Slight phase separation in PEI or DETA-PEGDA(400, 600, 1000) reaction systems in EtOH would make the gels soft.

The aza-Michael addition reaction of PEI or DETA and PEGDA200 in EtOH yielded the gels and porous polymers depended on the reaction conditions. Figure 5 shows the production diagram of PEI-PEGDA200 reaction systems in EtOH at 20 °C. The porous polymers were obtained under the feed molar ratios of PEI/DEGDA200 in the range of 2/12–2/14 and monomer concentrations in the range of 20–25 wt %. The feed molar ratio of PEI/DEGDA200 of Case 2 corresponds to 2/7.9. That means the porous polymers were obtained under conditions in which the molar concentrations of acrylate groups were higher than that of active hydrogens. The unreacted acrylate groups of PEGDA200 would form dangling chains in the reaction system, which would be usable to yield porous polymers.

Figure 6 shows the production diagram of DETA-PEGDA200 systems in EtOH, monomer concentration: 20 wt %. The porous polymers were obtained under the wide range of reaction conditions. The equivalent molar feed of active hydrogen to acrylate, feed molar ratio of DETA/PEGDA200: 2/5 corresponding to Case 2, must be suitable in this reaction system to obtain the porous polymers at a wide range of polymerization temperatures. The effect of the monomer concentration and polymerization temperature on the production state of the DETA-PEGDA200 system was investigated in the reactions of Case 2, as shown in Figure 7. The reactions with low monomer concentration, 15 wt%, yielded the porous polymer at a wide range of reaction temperatures, from 20 to 50 °C. By contrast, the reactions with high monomer concentrations, 25–30 wt%, yielded the porous polymers at low reaction temperatures, 10–15 °C. One explanation of the results is that the polymerization (gel formation) rate tends to be much higher than the phase separation rate in the reactions with high monomer concentrations owing to the high crosslinking density in the reaction system, and low reaction temperatures should be necessary to induce the phase separation by reducing the solubility of the polymer network.

The surface structure of the PEI or DETA-PEGDA200 porous polymers prepared in EtOH was observed by SEM. Figure 8 shows the SEM images of the PEI-PEGDA200 (a, b, c) and DETA-PEGDA200 (d, e, f) dry porous polymers obtained from the reaction systems with 20, 25, and 30 wt% of the monomer concentrations. The morphology of the polymers obtained from the reaction systems with 20 wt% monomer concentration was formed by connected spheres about 10 μm in both reaction systems, as shown in Figure 8a,d. The increase of the monomer concentration in the reaction systems, 25 wt%, decreased the size of the spheres, as shown in Figure 8b,e. Co-continuous monolithic structure was observed in the polymers obtained from the reaction systems with 30 wt % of the monomer concentration, as shown in Figure 8c,f. These structures should be induced by the spinodal decomposition during the network formation in the reaction systems [29,30,31,32]. In the case of the high monomer concentration reaction (30 wt%), the phase separation was fixed at the early stage of the spinodal decomposition of the co-continuous monolithic structure owing to the high polymerization rate with the high crosslinking density in the reaction system. The structure of connected spheres in Figure 8a,b,d,e would be formed at the late stage of the spinodal decomposition owing to the low polymerization rate in the reactions with a lower monomer concentration (15 and 20 wt %).

The porosity of the dry porous polymer was determined by the following equation:*Porosity* [−] = 1−𝑊/𝜌𝑉(1)
where *W*, *ρ*, and *V* are weight (g), true density (g/cm^3^), and volume (cm^3^) of the porous polymer, respectively. The porosity of the dry porous polymers ranged from about 65 to 70%, and decreased with the increase in the monomer concentration in the reaction system, as summarized in Table 4. Porous polymers with PEI and DETA showed similar values of porosity. The specific surface area of the porous polymers was determined by the BET method (Appendix A), and the results are summarized in Table 4. The values were relatively low owing to the not micro-porous, but macro-porous structure of the porous polymers. The DETA-PEGDA200 porous polymers showed a larger surface area than the PEI-PEGDA200 porous polymers. One explanation of the result is that the uneven structure on the skeleton of the DETA-PEGDA200 porous polymer, as observed in Figure 8f, would increase the specific surface area.

Mechanical properties of the PEI and DETA-PEGDA200 dry porous polymers, which were obtained from the reactions with the feed molar ratio of PEI/PEGDA200: 2/14 or DETA/PEGDA200: 2/5 (Case 2) in EtOH, were investigated by the compression test. Stress–strain curves of these porous polymers are shown in Figure 9, and the results are summarized in Table 4. All the porous polymers were soft and flexible, and unbreakable under the pressure of 50 N. The compressed porous polymers quickly returned to the original shape when the pressure was released (Appendix A). Young’s modulus of the porous polymer increased with the increase in the monomer concentration in the reaction system owing to the increase of the true density (decrease of the porosity). The Young’s modulus values of the porous polymers with PEI were lower than those with DETA. The branched PEI structure should induce soft and flexible features of the porous polymers.

The dried porous polymer absorbs various solvents owing to the high affinity of the PEG linkage to the solvents. Figure 10 shows the absorption capacity of PEI or DETA-PEGDA200 porous polymers obtained from the reaction systems in EtOH with 20 wt % monomers. The polymers absorbed the solvents listed in Figure 10 and gained 100–350% in volume based on the original size. The solvents were absorbed by soaking to vacant space and swelling of the spheres in the porous polymers. The volume gain of the porous polymers by the absorption is induced by swelling of the spheres. The swelling ratio should be affected by the affinity between the polymer network and the solvents. The affinity between PEG and the solvent was quantitatively evaluated by Hansen solubility parameter (Hansen distance: R_a_ = {4*(dD_1_ − dD_2_)^2^ + (dP_1_ − dP_2_)^2^ + (dH_1_ − dH_2_)^2^}^1/2^, where 1: PEG; 2: solvent; and dD, dP, and dH are the energy from distribution forces, dipolar intermolecular forces, and hydrogen bond between molecules, respectively) [33]. Hansen distance values of PEG-DMSO, PEG-dichloromethane, and PEG-chloroform are relatively smaller than other solvents, indicating high affinity between PEG and these solvents (Appendix A). The high affinity between the polymer network and DMSO, dichloromethane, or chloroform should induce a high swelling ratio of the porous polymers.

## 4. Conclusions

The network polymers with a PEG unit were successfully synthesized by ring opening addition reactions of conventional multi-functional amines, PEI or DETA, with PEGDE in H_2_O at room temperature without a catalyst, or in DMSO at 90 °C using PPh_3_ as the catalyst. The network polymers were also synthesized by the aza-Michael addition reaction of PEI or DETA with PEGDA in DMSO or EtOH at room temperature. The feed ratio of amines to epoxy or acrylate group, structure of amine monomers, molecular weight of PEGDE or PEGDA, and feature of the solvent affected the gel formation time and mechanical properties of the resultant gels. The increase of the PEGDE or PEGDA feed ratio (Case 2) induced a long gel formation time owing to the steric hindrance of secondary amines formed by the addition reaction of a primary amine and one epoxy or acrylate group. The gels formed by higher feed ratio of PEGDE (Case 2) showed a higher Young’s modulus owing to the high crosslinking density. By contrast, the gels with PEGDA showed opposite results owing to the low reaction conversions in the reactions of Case 2. The increase of the molecular weight in PEGDE or PEGDA increased the gel formation time and decreased the Young’s modulus of the gels owing to the low crosslinking density under the same monomer concentration. The reaction systems with PEI showed a longer gel formation time and yielded gels with a higher Young’s modulus than the corresponding reaction systems and gels with DETA. These results indicate inter-penetration of the polymer networks derived from the specific structure of PEI, which would play a role of the physical crosslinking points.

The reaction of PEI or DETA with low molecular weight PEGDA200 in EtOH under specific conditions yielded porous polymers induced by phase separation during the network formation. The morphology of the porous polymers showed connected spheres or a co-continuous monolithic structure depending on the reaction conditions. The porous polymers showed flexible features and were unbreakable by the compression. The porous polymers absorbed various solvents owing to the high affinity with a PEG unit in the polymer network.

As mentioned above, the addition reactions of the multifunctional-amines, PEI or DETA, and PEGDE or PEGDA based on the joint and linker concept must be usable methods to synthesize the network polymers containing a PEG unit. We are studying the characteristic network structure in the porous polymers, especially the formation process of the co-continuous structure, and applications such as column, support of conductive material, separator of battery, and scaffold of cell cultivation, among others, and the results will be reported elsewhere.

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
