# Peer review of "Synthesis of Network Polymers by Means of Addition Reactions of Multifunctional-Amine and Poly(ethylene glycol) Diglycidyl Ether or Diacrylate Compounds"

_polymers, 2020, doi:10.3390/polym12092047_

Round 1
Reviewer 1 Report
This is an interesting gel formation study - basing on system of broad practical importance. Although long-known building blocks were used to make rather innovative and interesting gels (including hydrogels) either utilizing amine-epoxy or aza-Michael addition reactions. The work represents a useful contribution to the knowledge of crosslinking of these systems and knowledge on final properties. The article (especially the phase-separation studies) also shows how rich is the variety of parameters in studying crosslinked systems.
The article offers learning and I find it suitable for polymers - after some corrections and complementing.
I have couple of specific points to follow:
Page 2, line 65: instead of third amine better tertiary amine
Paragraph 2.2.1.
Instead of "ample tube" better "ampoule"
lines 113, 121 Instead of "sealed by burning off" better just "sealed" - this is normally used in connection to sealing glass ampoules
Page 4, line 136: Mechanical properties: was the compression test done in solvent or with samples out of the solvent
Using BET for surface area of these MACROporous samples: can you justify the choice of the method? BET is known to have limits when the pores are large, i.e. > 1 um - macroporous samples - communicating pores - are typically characterized with mercury porosimetry.
The specific surface areas are quite low - I wonder if such values are realistic. Pls. compare with literature of similar systems.
Page 5, line 158 - how was the critical gelation concentration determined?
Page 5, line 175, instead of DATE write DETA
Page 6, line 180 Instead of DEGDE write PEDGE
Table 1: the conversion values - are these molar % of epoxy groups? How precisely is the conv. determined - is the decimal point meaningful?
Pls explain why the conversion is determined for solution of 3 wt.% of monomers while the gelation time is determined for your 30%-wt. systems?
Although, the determination of conversion should be explained: to which reaction state are the numbers - values of conversion in the Tables 1, 3 related? Always after 24 hours of reaction time? Such values are not much illustrative. Why not to present reaction half-life time of the "real" systems and critical conversion instead?
Tables 2 and others relevant: pls call the variables rather "stress at break", "strain at break"
Titles of parts 3.1. and 3.2. are not much different - but the reaction mechanisms are much different.
Figure 10: the font size - especially solvent names - is very small
Author Response
For the comments of Reviewer #1
- The words in lines 65, 113, and 121, and paragraph 2.2.1 have been corrected according to suggestion of the reviewer.
- The compression test of “wet gel” or “dried porous polymer“ was conducted. These explanations have been added in the revised manuscript in line 136.
- As the reviewer pointed out, characterization with mercury porosimetry must be better for the macro porous polymers. Unfortunately, we do not have the mercury porosimeter. We applied available BET method to evaluate qualitatively for reference.
We reported synthesis and characterization of some porous polymers based on the joint and linker concept in refs 5, 6, and 7. The surface areas of the previous reported polymer were also low. This would be a feature of the porous polymers synthesized by the joint and linker concept. Further theoretical studies about this point are under investigation, and the results will be reported elsewhere.
- The critical gelation concentration, the lowest monomer concentration which yields the gel, was determined by the reactions with various monomer concentrations. The critical gelation concentrations of the reactions were about 5 wt%. But what we would like to say here is that the measurement was conducted in the low monomer concentration, which was not enough to form the gels. The explanation has been added in the revised manuscript.
- The words in lines 175, and 180 have been corrected accordingly.
- Reaction conversions, conversion of the epoxy group, of the model reaction systems in H2O or DMSO with low monomer concentration (3 wt%), which was not enough to form the gels, were quantitatively evaluated from the peak intensity ratio of CH of epoxy group detected at 2.7-2.8 ppm and methylene of PEG unit detected at 3.5 ppm in 1H NMR spectroscopy for references of those in gels.
The explanation has been added in the revised manuscript.
- The 1H NMR spectroscopy of the gels with monomer concentration of 30 wt% was impossible due to the broadening of the peaks. Reaction conversions of the model reaction systems in H2O or DMSO with low monomer concentration (3 wt%), which was not enough to form the gels, were quantitatively evaluated for references of those in gels.
The explanation has been added in the revised version.
- The reaction conversions of PEGDA tended to be larger than those of PEGDE. That should induce shorter gelation times of the reaction systems with PEGDA.
We continued all the reactions of syntheses for 24 h, even after the gelation, to attain the equilibrium swelling of the gels. The reaction may proceed after the gelation. We used the gels for compression test obtained by the reaction for 24 h. The reason why we measured the reaction conversions of the model samples after the 24 h.
- The words of “breaking stress” and “breaking strain” have been changed to “stress at break” and “strain at break” in the revised manuscript.
- The titles have been revised accordingly
- Large font size has been used in Figure 10 in the revised manuscript.
Reviewer 2 Report
- Scheme 1 need to be more clear as it seems that there are two reactions at once.
- 98 five NH moieties should be three
- One should mention the role ofPPH3 as a catalyst
- 150 The reaction in EtOH did not gel
- FTIR should have numbering of the arrows.
- Quality of the NMR spectrum shoulded be improved, include dessignation of all peaks including solvent peaks
Author Response
For the comments of Reviewer #2
- Scheme 1 has been corrected accordingly.
- DETA has two primary amines, NH2 group, and one secondary amine, NH group. That means five NH moieties or three amine groups in one DETA molecule.
- Activation of epoxy group.
- The sentence has been corrected accordingly.
- We numbered allows in the FT-IR spectra.
- The NMR spectra have been modified in the revised manuscript.
Round 2
Reviewer 1 Report
The paper still contains many language mistakes. I strongly recommend language revision: style and grammar.
There are still unclear points. These have to be cleared out before presenting this work to public.
1/ Part 2.2. It is not lucky to say that DETA contains FIVE NH moieties. Although I guess what you mean, it is not true: it virtually contains only three NH moieties. You can say it contains FIVE actibe hydrogens – active in reaction with epoxy group. The explanation of stoichiometry of the studied chemistries is a crucial point of this work. Please check and reformulate lines 101-103 of page 3 – explanation of stoichiometry for the aza-Michael cases. You could mention – graphically - that the Scheme 3 is actually a general scheme for the both studied chemistries.
2/ Part 3
The description of the both FT-IR figures (Fig1 and 3) is confusing. The peaks are described wrongly in the text. Please check: for example, in the Figure 1, the peak (i) is supposed to appear at 800 cm-1 according to the description above but it is marked at ~1600 cm-1 in the Figure 1. The other peaks and the Figure 2 are confused also. The best is to assign the peaks and describe the groups perhaps graphically directly in the Figures?
page 5, line 167: “Low reactivity of secondary amine which is fomed by the reaction of a primary amine... etc”: Add reference that the reactivity of sec.amine in water is lower than the reactivity or prim. amine. Normally, if no proton donor is present (here water is present) and in the case of aliphatic amines, the reaction rate of sec.amine with epoxy group is catalyzed by the reaction product of the reaction or prim. amine with epoxy group (i.e. with OH groups). So, because you state the opposite, this point must be supported by solid argument, experiment or a reference. The difference in conversion can be for other reasons, for example phase separation.
Phase separation in the 3% wt. solutions referred to in the Tables 1 and 2 is very likely. Were the studied solutions always clear and not phase separated?
In general, comparing behavior of diluted non-gelling samples with the 10 times concentrated and gelling samples is not a very compelling concept as the solution behavior at different concentrations may be very diverse. Therefore, the conversion column in the both mentioned Tables is quite confusing. Conversion of expoxy group/double bond could be measured for the gelling mixtures at least up to the gel point by FT-IR. In fact, it is hard to find a sense in such a comparison next to the gelation times.
Gelation times. When measured from viscosity uptake can be misleading and should be taken only informatively. Therefore, it should be made very clear that the gelation times from viscosity measurements are rough values, they are rather pot-life times. The gelation time may be biased by the formation of many hydrogen bonds in the systems, in other words, the mixtures whose viscosity increases three or five times can still be far from chemical gel point. The urea and epoxy systems are known for these “physical gelation” effects. The gel point should be determined from solubility in a very good solvent (maybe chloroform in this case?). Thus the discussion on gelation times compared with the conversion – measured however with differently diluted smaples! – should be re-written and its meaning explained. Definitely, putting the conversion values after 24 hours next to the gel times is quite misleading and wrong to make links between the two values.
page 7, line 220: “the reaction ...geled (should be gelled) but even if, this is not a good expression. It should be said that “the reaction system gelled”.
The same expression can be found later, too.
page 8, line 254
PLs explain why the higher crosslink-density should induce higher reaction rate: there is no general expectation valid for such a situation. It is quite opposite: highly crosslinked system may transfer to its glassy region and there the reaction constant falls to low values or high crosslink density may result in phase separation, this can also cause transition to glassy state and slowing down the reaction.
Mechanical properties. Explain what was the state of the samples at which the mechanical properties were studied – were the samples swollen to equilibrium in reaction solvent? What was the purpose to study dry porous samples with mechanical tests?
Page 13, line 347
The term “globule” is typically reserved for the coiled state of macromolecules (single chains). I suggest to replace it for a term “sphere”.
Figure 10.
The swelling should be expressed volume-vise. The weight picture is misleading – as you say – because of different specific gravity of the solvents. By the way, when recalculated to volume swelling the values are not much the same. The swelling part deserves some explanation of the observed trend.
Author Response
Thank you very much for your letter of July 29 and for the referees’ comments concerning our manuscript entitled "Synthesis of network polymers by means of addition reaction of multifunctional-amine and poly(ethylene glycol) diglycidyl ether or diacrylate compounds" (polymers-855284). We found the referees’ comments most helpful and have revised the manuscript accordingly. Revised and added portions are written in blue.
The replies for the comments are listed below.
For the comments of Reviewer #1
Native check of the manuscript is in progress.
1/Part 2.2.
The explanation of “feed ratio” of monomers in the experiments have corrected accordingly. We also revised Scheme 3.
2/Part3
Allows in the Figures 1 and 3 were miss numbered. The number in the figures have been corrected in the revised manuscript ((i) à (iii), (iii) à (i)).
Page 5, line 167:
We deleted all the data and discussions based on the NMR analysis of diluted model samples to avoid the confusing of the readers.
We could not find definite difference in the conversions in the PEI-PEGDE400/DMSO reaction systems of Case 1 and Case 2 by FT-IR spectroscopy of the gels. The result shows that the reaction conversions of theses gels were almost the same. In the reactions of Case 2 would procced two steps, completion of reaction of primary amine and an epoxy following secondary amine and another epoxy, as reported in curing process of diamine and bisphenol A type diepoxy [Yamasaki, H.; Morita, S. Epoxy curing reaction studied by using two-dimensional correlation infrared and near-infrared spectroscopy. J. Appl. Polym. Sci. 2011, 119, 871-881.]. The longer gel formation time of Case 2 would be caused by the two step reactions of amine and epoxy groups. Steric hindered secondary amine formed by a reaction of primary amine and one epoxy would also be possible.
The longer gel formation time of Case 2 in PEI-PEGDA400/DMSO system would be caused by steric hindered proton transfer of the secondary amine formed by a reaction of primary amine and acrylate, as computational studied an aza-Michael addition reaction [Desmet, G.B.; D’hooge, D. R.; Omurtag, P.S.; Espeel, P.; Marin, G.B.; Du Prez, F.E.; Reyniers, M.F. Quantitative First-Principles Kinetic Modeling of the Aza-Michael Addition to Acrylates in Polar Pprotic Solvents. J. Org. Chem. 2016, 81, 12291-12302].
These sentences and references have been added in the revised manuscript.
The term of “Gelation time” has been changed to “gel formation time”. An infection point of the profile of the viscosity was defined as the gel formation time. The gel formation time does not mean theoretical gelation time. These explanations were added in the revised manuscript.
Page 7, line 220:
All the expressions have been changed accordingly.
Page 8, line 254
We have changed the pointed part to following.
“The higher molar concentration of the acrylate (and amine) group in the reaction system with low molecular weight PEGDA200 yielded the polymer network with high crosslinking density, which should cause the short gel formation time in the homogeneous phase.”
In the reaction system, clear homogeneous gel was formed without phase separation. We think the transition from homogenous state to glassy occurs may not occur.
Mechanical properties of the porous polymers were tested at dry state. The morphology, observed in SEM, of porous polymer should affect the mechanical properties. SEM images were observed at dried state. We think it must be worthwhile to investigate the mechanical properties of porous polymer at dry state to discuss the relationship between the morphology in SEM and the mechanical properties.
Page 13, line 347
The term has been replaced accordingly.
Figure 10
The absorption capacity is evaluated in revised Figure 10. Volume gain of the systems was discussed based on affinity between PEG and solvents using Hansen solubility parameter.

Round 3
Reviewer 1 Report
It was a pleasure to read your manuscript V3 as you constructively followed my comments. I believe it promoted the clarity of the message.
Content-wise, the manuscript is now ready for publication.
Language-wise: minor grammar and style corrections should be made.
For example:
Page 5/ line 150 did not gel instead of did not gelled
Page 5/ line 164 an inflection point instead of an infection point
Page 5/ line 172 these gels instead of theses gels
Page 5/ line 173 proceed instead of procced
Page 6/ lines 176,177 better just present tense "is" instead of conditional tense "would" (why conditional tense?)
Page 13/ line 386 th sentence is incomplete ...The swelling should be the main ...reason...?
I did not detect all typos etc, the ms needs one more careful read by the authors.
Author Response
Thank you very much for your letter of Aug. 10 and for the referees’ comments concerning our manuscript entitled "Synthesis of network polymers by means of addition reaction of multifunctional-amine and poly(ethylene glycol) diglycidyl ether or diacrylate compounds" (polymers-855284). We have studied their comments carefully and have made corrections which we hope to meet with their approval. Revised and added portions are written in green.
The replies for the comments are listed below.
For the comments of Reviewer #1
All the words and expressions have corrected accordingly. Native check of the manuscript is in progress and will be finished soon.
What we would like to say here is that “The volume gain of the porous polymers by the absorption is induced by swelling of the spheres.”
The explanation has replaced in the revised manuscript.